# OpenReview forum: "Parameter-Efficient Tuning Helps Language Model Alignment"
_ICLR.cc/2024/Conference — ICLR 2024 Conference Withdrawn Submission_

### Official Review · Reviewer_HEu9 · 2023-10-25

**Soundness:** 2 fair
**Presentation:** 3 good
**Contribution:** 2 fair
**Rating:** 5
**Confidence:** 4

**Summary:**

The paper explores how to align LLMs with human preference. It presents empirical results on two public datasets (one conversational dialog one from Anthropic, and one on the summarisation of reddit posts from OpenAI) which each have two targets, one a "good" and a "bad" response.
The paper talks about how RLHF and the more recent DPO formulation are lacking in that they only allow binary preferences. It proposes to do alignment instead via control codes, ie a prompt which is either static text encoded per the LLMs tokenisation+embedding, or learnt embedding of the same dimension as the LLMs embeddings. There has been lots of related work, e.g. the non-cited CTRL paper, which uses control codes for controllable generation (e.g. to control style, or sentiment of generated text), however this paper is novel I believe in looking at this for aligning an LLM produced via unsupervised learning.

The paper however is limited in that it blends in param efficiency of the prompt, which IMHO appears to be a different topic entirely, and the results on binary pref data are not convincingly better than the more rigorous DPO, nor are results given on non-binary preference which is purported to be a benefit of the proposed method.

**Strengths:**

* Results against public datasets are presented, with comparisons against some existing alignment baselines, notably DPO.
* Ablation study showing impact of including either just the soft-prompt learning, or just the further LLM fine-tuning (given a fixed soft/static prompt).

**Weaknesses:**

* The paper blends topics, without good justification. It is focussed on alignment, and presents a valid question on whether alignment can be achieved via control codes (static, or trained, ie soft-prompt-learning). However it introduces parameter efficiency, and IMHO I see no rationale for how this relates. The question of how to align an LLM can be addressed separately to requiring the learnt control codes to come from LORA adapted models or otherwise.
* Despite criticising some existing alignment works for being restricted to only optimising against binary ratings, the paper does not present any results on using control code based alignment to power non-binary preferences.
* Minor suggestion: Equation 2 is literally the same as equation 1. It's just the perspective one takes when looking at it (optimise LLM params, or optimise prompt params). This could be better written, and without the use of the duplicated equation.
* The results don't indicate that the method is reliably better than DPO. Most comparisons are given against the weaker CoH.

**Questions:**

* Why not plot win rates in Fig 3 rather than deltas?
* Apologies if I missed this detail -- are the same 128 points used for eval as was done by the DPO paper? How where these chosen otherwise?

---

### Official Review · Reviewer_z7DA · 2023-10-29

**Soundness:** 2 fair
**Presentation:** 3 good
**Contribution:** 2 fair
**Rating:** 3
**Confidence:** 4

**Summary:**

The paper proposes MEET, a method to train a LLM to generate "good" and "bad" answers to a given question / task by conditioning the model computation with an adapter (LoRA or Soft Prompt). To do so, they adopt a two-step training procedure. First, they train a "good control adapter" and a "bad control adapter" on good answers and bad answers respectively while keeping the base LM fixed, then they fine-tune both the control adapters and the base model. The authors show that this two step procedure is important to achieve gains over the Chain of Hindsight baseline (basically a baseline where control adapters is just a handcrafted prompt "A good/bad conversation is:") and DPO on two datasets OpenAI Summary and HH-RLHF from Anthropic.

**Strengths:**

- The paper is well-written, the details of the experimental setting are clear.
- The two-stage training procedure is interesting and its importance is validated by the ablation study.
- Results seem to suggest that the two-step optimization method delivers gains w.r.t. DPO.

**Weaknesses:**

- It feels like the authors are a bit confused on where the novelty of their paper really lies, they seem to suggest that it is in using adapters to control generation, but imho, the interesting bit is more on the two-step training procedure that guarantees information is captured by the adapters and thus they are not "information-starved" by the full LM fine-tuning (easy to fix)

- The more problematic bit is that authors' confusion seems to have affected the overall experimental methodology; for example, the authors seem to tie their method to the specific loss function used (i.e. MLE) and compare to DPO, while their method can be used on top of DPO. Moreover, the baselines numbers are a bit concerning and some important baselines are missing (overall harder to fix)

**Questions:**

About novelty:

The paper proposes to learn "attributes" conditional models with adapters, which have been proposed in https://arxiv.org/pdf/2302.08453.pdf for diffusion models for example. So, here, the novelty might reside in 1/ applying this general idea to textual generation tasks and 2/ the two-stage training approach proposed to train these adapters. The current stance of the paper is that the main novelty is to apply LoRA adapters for generation instead of hard prompts. I feel like 2/ is a more interesting and impactful contribution but currently it is a bit understated in the paper, so it feels like it should be the central focus of the paper. I feel like the paper can be an interesting set of experiments showing that the two-stage approach prevent adapters from being "information-starved" from full model fine-tuning.

About experiments:

Confusion about the contributions seem to appear in Section 3, where the authors tie their method to MLE loss (1) and (2) and compare in the experiments with a DPO baseline. This is a bit surprising to me given that their method can be deployed on top of DPO, i.e. Eq (1) and (2) can use DPO instead of MLE (to train each good and bad expert), so I am not sure why DPO would be a baseline in the experiments. On the contrary, I would have expected to see two versions of their method in the experiments: with Eq. (1) and (2) using DPO (MEET-DPO) and Eq. (1) and (2) using MLE (MEET-MLE).

From all experiments, one straightforward baseline is missing in addition to CoH: SFT -- which just trains on positive data.

Similarly, for MEET-MLE, what is the impact of integrating negative data? i.e. what is the gap between MEET-SFT, which just trains the controllable adapter of positive data and MEET-MLE, which trains on both positive and negative data with Eq. 2?

In the first dataset, DPO underperforms CoH on OpenAI/Summary dataset. The fact that DPO underperforms CoH on this dataset is a bit suspicious. Did you tune the \beta parameter for DPO on both datasets ?

How do you do cross-validation in these two datasets? Are you searching for the best HPs for each method on the validation set?

Taken together, your results currently show that DPO is useless in these two datasets and severely underperform MLE training with MEET. I am not sure this result can be published without further ablations and baselines as I suggest above, especially it appears to me that MEET can be further improved with DPO training.

Please, do not consider my score as final, I am willing to increase the score substantially if the authors can give answers to my questions.

---

### Official Review · Reviewer_Q2P6 · 2023-10-29

**Soundness:** 2 fair
**Presentation:** 2 fair
**Contribution:** 2 fair
**Rating:** 3
**Confidence:** 4

**Summary:**

This paper proposes to use parameter-efficient tuning (e.g., prompting tuning and low-rank adaptation) to optimize control tokens
and then fine-tune models for controllable generations. The MEET aims to improve the quality of control tokens, thus improving controllable generation quality consistently by an apparent margin on two datasets.

**Strengths:**

1. This paper studies a parameter-efficient way to improve the language alignment. It is an interesting direction to explore.

2. It studies several aspects of the proposed method such as prompt length, rank, and temperature.

**Weaknesses:**

1. This paper conducted several experiments. However, I don't think the baselines the paper compares with are sufficient. Several works focus on a similar idea about incorporating the reward into text learning, such as RLPrompt [1] and AutoPrompt [2]. Those should become the baselines to compare the method proposed in the paper. Also, For controllable text generation, there is an interesting direction to utilize the diffusion process, such as the Diffusion-LM [3]. However, none of these are included and compared in the paper. Thus, I am not convinced with the experimental results shown in the paper.

2. The performance of the proposed method does not show enough improvements compared to the baseline mentioned in the paper. It highly correlates to the hyperparameter settings. It would be good to include the detailed ablations of those hyperparameters.

3. The proposed method seems to be not novel. We know the impact of LoRA, and the proposed method seems just a direct implementation of the LoRa with parameter-efficient tunning with some specific designs. Could authors provide more justification about the novelty of the proposed methods?

4. For the ablation section, what would be the efficiency comparison between the proposed method and the baselines? Such as the running time and computation latency.



[1] Rlprompt: Optimizing discrete text prompts with reinforcement learning

[2] AutoPrompt: Eliciting Knowledge from Language Models with Automatically Generated Prompts

[3] Diffusion-LM Improves Controllable Text Generation

**Questions:**

Please refer to the Weaknesses section.

---

### Official Review · Reviewer_Lxvm · 2023-11-01

**Soundness:** 3 good
**Presentation:** 2 fair
**Contribution:** 3 good
**Rating:** 6
**Confidence:** 3

**Summary:**

This paper considers the use of parameter-efficient fine-tuning techniques for alignment. Specifically, they consider the task of generating control tokens (via prompt tuning) and subsequently fine-tuning the model with these controls tokens (via LoRA). Across two benchmarks, the work finds that this join technique improves upon representative prior work of DPO for one benchmark.

**Strengths:**

1. The evaluation is thorough for the benchmarks considered, with 2 different evaluation metrics and ablation studies
2. The problem of controllable generation is important, allowing one to control model generation at inference time

**Weaknesses:**

1. One key ablation that is missing is doing stage 2 only (skipping the control token optimization) but starting with the CoH control tokens (or not even optimizing the CoH tokens at all). This would really elucidate the role of prefix optimization since if it is subsumed by CoH, it is not important that it is parameter-efficient (which is a central claim to the paper).

2. Training soft prompts before fine-tuning the model has been studied by the prior work of [promot](https://arxiv.org/abs/2211.00635) which finds similar performance improvements on the task of summarization.

3. The paper title is a little too general. Though the methods use PEFT, this is not essential to any of the results since the primary contribution is for two-stage optimization and only for the application controllable generation. If accepted, I would suggest updating the title to better reflect both of these specific attributes.

**Questions:**

1. Is there any intuition for why DPO performs better?

2. In my first read of the paper, I got confused for a long time with Section 3.3, thinking that the second stage was LoRA while the first stage was prompt tuning. Is it possible to better clarify that there is eventually full fine-tuning, and LoRA/prompt tuning is interchangeable as a choice for the first step?